# Bone Fusion Morphology after Circumferential Minimally Invasive Spine Surgery Using Lateral Lumbar Interbody Fusion and Percutaneous Pedicle Screws without Bone Grafting in the Thoracic Spine: A Retrospective Study

**DOI:** 10.3390/medicina58040496

**Published:** 2022-03-30

**Authors:** Masayuki Ishihara, Shinichirou Taniguchi, Koki Kawashima, Takashi Adachi, Masaaki Paku, Yoichi Tani, Muneharu Ando, Takanori Saito

**Affiliations:** Department of Orthopedic Surgery, Kansai Medical University, Osaka 573-1191, Japan; tanigucs@takii.kmu.ac.jp (S.T.); chelseajoesheva@gmail.com (K.K.); adachita@hirakata.kmu.ac.jp (T.A.); kmu.orthopaedics.pak@gmail.com (M.P.); taniyoic@gmail.com (Y.T.); mando@gaia.eonet.ne.jp (M.A.); saitot@takii.kmu.ac.jp (T.S.)

**Keywords:** adult spinal deformity, circumferential minimally invasive surgery, lateral lumbar interbody fusion, percutaneous pedicle screw, bone fusion process, bone fusion morphology, spontaneous bone fusion, without bone grafting

## Abstract

*Background and Objectives*: This study aimed to investigate the process and morphology of thoracic and lumbosacral bone fusion in patients with adult spinal deformity (ASD) who underwent circumferential minimally invasive spine surgery (CMIS) by lateral lumbar interbody fusion (LLIF) and percutaneous pedicle screws (PPSs) without bone grafting in the thoracic spine and who have risk factors for bone fusion failure in the thoracic spine. *Materials and Methods*: This retrospective study included 61 patients with spinal deformities (46 women and 15 men) who underwent CMIS with LLIF and PPSs at our hospital after 2016 and completed a 3-year postoperative follow-up. The rate and morphology of bone fusion and rod fracture rate in the thoracic and lumbosacral vertebrae were evaluated. Patients were divided into the thoracic spine spontaneous bone fusion group and the bone fusion failure group. The data of various spinopelvic parameters and the incidence of complications were compared. The vertebral body conditions in the thoracic spine were classified as less degenerative (type N), osteophyte (type O), and diffuse idiopathic skeletal hyperostosis (DISH) (type D). *Results*: After three postoperative years, the bone fusion rates were 54%, 95%, and 89% for the thoracic, lumbar, and lumbosacral spine, respectively. Screw loosening in the thoracic vertebrae was significantly higher in the bone fusion failure group, while no significant differences were observed in the spinopelvic parameters, Oswestry Disability Index (ODI), and rate of proximal junctional kyphosis and rod fractures. Type N vertebral body condition and screw loosening were identified as risk factors for spontaneous bone fusion failure in the thoracic spine. *Conclusion*: This study indicated that spontaneous bone fusion is likely to be obtained without screw loosening, and even if bone fusion is not obtained, there is no effect on clinical results with the mid-term (3-year) results of CMIS without bone grafting in the thoracic spine.

## 1. Introduction

As there is an unprecedented rise in population aging, the number of patients with adult spinal deformities (ASD) has been increasing steadily. Correction surgery has become widespread since Schwab et al. reported target alignment procedures [1]. However, complication rates of 8.4–68% were reported in association with such correction surgeries, while surgical invasion also remains a concern [2,3]. ASD is usually caused by the degeneration and/or deformation of the intervertebral disc, and its primary treatment is the surgical release and correction at the intervertebral disc level. Haque et al. (2014) and Park et al. (2015) reported the usefulness of lateral lumbar interbody fusion (LLIF) for the treatment of ASD [4,5]. Furthermore, circumferential minimally invasive surgery (CMIS) with LLIF and percutaneous pedicle screw (PPS) fixation procedures are gradually increasing in prominence [4,6]. The extent of minimal invasiveness of CMIS for ASD is clear compared to that of conventional open surgery and the hybrid method that combines LLIF with posterior open surgery [4,5]. Furthermore, some reports demonstrated the preventive effect of proximal junctional kyphosis (PJK), which is achieved by minimizing the detachment of the posterior soft tissue with PPSs [6,7,8]. However, bone fusion failure due to insufficient surface of the bone graft is the most serious problem in CMIS for ASD [9]. In our institution, we perform LLIF at the lumbar level, mini-open transforaminal lumbar interbody fusion (TLIF) for L5/S1, and posterior fusion with PPS from the upper instrumented vertebra (UIV) to the lower instrumented vertebra (LIV). Hence, bone grafts were not performed in the thoracic spine, as concerns remain regarding implant-related complications associated with bone fusion failure. In contrast, spontaneous bone fusion was observed in some of our cases in the thoracic spine without bone grafting. Nevertheless, no previous reports have demonstrated spontaneous bone fusion in the thoracic spine. Therefore, this retrospective study was designed to investigate the rate and morphology of bone fusion in the thoracic and lumbosacral spine, the incidence of implant-related complications associated with bone fusion failure, and the risk factors of thoracic spontaneous bone fusion failure in CMIS using LLIF and PPSs without bone grafting in the thoracic spine.

## 2. Materials and Methods

This study was approved by the Institutional Review Board of the Kansai Medical University Hospital (2020189). The study was conducted in accordance with the principles of the Declaration of Helsinki.

Patients with ASD who underwent corrective surgery using LLIF and PPSs between October 2016 and March 2018 and completed a 36-month postoperative follow-up were evaluated. The inclusion criteria were as follows: age > 50 years, PI-LL > 20°, PT > 20°, and fixation from the lower thoracic spine to the pelvis. Written informed consent was obtained from the patients prior to study participation and for the publication of this report and any accompanying images.

Patients with upper- and mid-thoracic vertebrae to pelvic fusion, spinal fusion below L1, a history of instrumentation surgery of two or more intervertebral discs, a history of three-column osteotomy, and insufficient radiographic data were excluded. The evaluation items were bone fusion rate, bone fusion morphology, and rod fracture rate in the thoracic spine, lumbar spine, and lumbosacral spine up to three years after surgery. Furthermore, the patients were divided into two groups based on achieving bone fusion in the thoracic spine: the spontaneous bone fusion group (Group U) and the bone fusion failure group (Group NU). Among these groups, the following factors were examined: preoperative vertebral body conditions, various spinopelvic parameters, the occurrence of complications (including screw loosening), PJK, rod fractures, Hounsfield units (HUs) of the UIV, and the Oswestry Disability Index (ODI).

Bone fusion morphology was classified into three types: posterior fusion (type P), interbody fusion (type I), and bridging (type B) (Figure 1). The vertebral body conditions in the thoracic spine were classified as less degenerative (type N), osteophyte (type O), and diffuse idiopathic skeletal hyperostosis (DISH) (type D) (Figure 2). Type N was defined as the condition in which disc height is maintained without osteophytes or vacuum phenomena.

### 2.1. Surgical Technique

Two-stage surgery was performed in all cases. First, LLIF was performed from L1/L2 to L4/5, and one week later, mini-open TLIF at L5/S1 and posterior fixation using the PPS procedure were performed. Bone grafting of the thoracic spine was not performed for any patient (Figure 3). The PPS system (Precept^®^ or Reline MAS^®^; NuVasive, Inc., San Diego, CA, USA) was used in all levels from the UIV to LIV. Extreme Lateral Interbody Fusion (XLIF^®^; NuVasive, Inc., San Diego, CA, USA) was performed, and 5.5-mm titanium alloy rods were used in all cases. Furthermore, an XLIF titanium 10° lordotic cage (Corent XL Ti^®^; NuVasive, Inc., San Diego, CA, USA) was used in all cases. The cage was filled with autologous ilium and a hydroxyapatite/collagen composite (Refit^®^; HOYA Technosurgical Co., Tokyo, Japan).

### 2.2. Radiological Evaluation

Various spinopelvic parameters were evaluated with the standing full-length lateral view (Figure 4). The following implant-related parameters were investigated with reference to the report by Ishihara et al. [10]: the kyphotic angle of the rod in the UIV-L1, represented as the rod kyphotic angle (RKA) and the angle between the PS and cranial endplate in the UIV (pedicle screw angle [PSA]) (Figure 4). The bone fusion rate, bone fusion morphology, and preoperative vertebral body conditions were evaluated by reconstruction of 3-dimensional computed tomography (3DCT) images. The HU value of the UIV was measured using computed tomography (CT). Based on the method described by Schreiber, HU was measured using circular regions of interest (ROIs) of the axial CT slice [11]. The largest range of cancellous bone, except the cortical bone, was the ROIs (Figure 5). Bone fusion was defined as the presence of complete intervertebral bridging or trabecular continuity within the cage and vertebral endplates on 3DCT images. PJK was defined based on the proximal junctional angle (PJA) between the caudal endplate of the UIV and the cranial endplate of two levels above the UIV. If the PJA ≥ 20° or the PJA increased by ≥20° than the preoperative PJA on plain radiographs obtained in a standing full-length lateral view [12], the patient was diagnosed with PJK.

### 2.3. Statistical Analysis

Radiographic and clinical parameters were analyzed using the Student’s *t*-test for continuous data and Fisher’s exact test for categorical data. Statistical significance was set at *p* < 0.05. All analyses were performed using JMP software (SAS Institute Inc., Cary, NC, USA).

## 3. Results

One hundred and nineteen patients were recruited, of whom 58 were excluded: seven underwent surgeries from the upper thoracic vertebrae to the ilium, 35 below the lumbar spine, seven with less than three years follow-up, and nine with insufficient radiological data). Thus, 61 patients (thoracic spine, 191 levels; lumbar spine, 242 levels; lumbosacral spine, 61 levels) were included in the final analysis (Figure 6); demographic and baseline data are shown in Table 1.

After three postoperative years, the bone fusion rate in the thoracic spine was 54% (new bone fusion, 42%; bone fusion, 12%). The bone fusion rates in the lumbar spine and lumbosacral spine were 95% and 89%, respectively (Table 2). Regarding the morphology of bone fusion, the ratio of type B was the highest in the thoracic, lumbar, and lumbosacral spines. In 37% of cases of lumbar bone fusion, both types B and I were confirmed to be mixed, and in 30%, both types B and P were confirmed to be mixed. In the lumbar and lumbosacral spines, type I was the second most common, whereas type P was the second most common in the thoracic spine (Table 2).

Regarding rates of complications, the incidence of screw loosening was 46%, of which 13% was present in UIV and UIV-1, and 33% was found in UIV. The incidence of PJK was 20%. Rod fractures were present in 15% of the lumbar spine cases, of which two cases were L2/3, three were L3/4, and four were L4/5; 13% of rod fractures occurred in the lumbosacral spine (L5/S1) (Table 1). No lesions were observed in the thoracic spine.

In one case, an RF after bone fusion was observed. Among the preoperative vertebral conditions in the thoracic spine, type N was present in 40 cases, type D was present in 11 cases, and type O was present in ten cases. The bone fusion rates were 35%, 91%, and 80%, respectively, with the bone fusion rate of type N being the lowest (Table 3).

Comparing the thoracic spines in Group U and Group NU, the proportion of type D was significantly higher, and the proportion of type N was significantly lower in Group U. Screw loosening was significantly higher in Group NU, but there was no significant difference in the occurrence of PJK. The total RF occurrence was significantly higher in Group NU (Table 4).

There were also no significant differences between the two groups in HU and reoperation rates due to PJK and RF (Table 4). Comparing various spinopelvic parameters between both groups, the preoperative thoracic kyphosis (TK) and PSA were significantly larger, and postoperative sagittal vertical axis (SVA) was smaller in Group U than in Group NU (TK, *p* = 0.006; post-SVA, *p* = 0.037; PSA, *p* = 0.025) (Table 5). The preoperative ODI was 35.2 points in Group U and 33.4 points in Group NU, and the postoperative ODI was 29.3 points in Group U and 27.1 points in Group NU, without any significant difference (preoperative, *p* = 0.231; postoperative, *p* = 0.312). Based on the results of the univariate analysis, five items (pre-TK, post-SVA, PSA, vertebral condition type N, and screw loosening) were independent variables, and spontaneous bone fusion failure in the thoracic spine was the dependent variable. Logistic regression analysis revealed that post-SVA, type N vertebral body condition, and screw loosening were risk factors for spontaneous bone fusion failure in the thoracic spine (Table 6).

Thereafter, we grouped the patients with and without screw loosening (Group SL and Group NSL, respectively). The various spinopelvic parameters, implant-related parameters, HUs, occurrence of complications, and revision rates were compared by univariate analysis among patients, and the risk factors for PS loosening were further analyzed by multivariate logistic regression analysis. There were no significant differences between the two groups in terms of age, sex, or HU in the UIV. In Group SL, type N was significantly higher, and in Group NSL, type D was significantly lower. The bone fusion rate in the thoracic spine was significantly lower in Group SL, but there was no significant difference in the incidence of mechanical complications, such as PJK and RF (Table 7).

Comparing various spinopelvic parameters between the two groups, the preoperative and postoperative PI and preoperative LL were significantly smaller, and preoperative TK and PSA were significantly larger in the NSL group than in the SL group (preoperative PI, *p* = 0.023; postoperative PI, *p* = 0.031; preoperative LL, *p* = 0.041; preoperative TK, *p* = 0.015; PSA, *p* < 0.001) (Table 8). Based on the results of the univariate analysis, six items (pre-PI, post-PI, pre-LL, pre-TK, PSA, and vertebral condition type N) were independent variables, and screw loosening was the dependent variable. Logistic regression analysis revealed that the PSA degree was a risk factor for screw loosening in the thoracic spine (Table 9).

Furthermore, receiver operating characteristic analysis of screw loosening using PSA revealed that the cutoff value for PSA was 15.3°, and the area under the curve was 0.731 (Table 10).

### 3.1. Case Study

#### Case Study 1

A 75-year-old man with degenerative kyphoscoliosis and severe back pain underwent CMIS with LLIF and PPSs. In the thoracic spine, the preoperative vertebral conditions were type O at T10/11 and type N at T11/12/L1. Two years after the surgery, type P bone fusion was observed in the thoracic spine. (Figure 7)

## 4. Discussion

In recent years, minimally invasive spinal fusion for various spinal diseases has become widespread [4,6,13,14]. In particular, the benefits of minimal invasiveness in correction surgery for ASD are great. On the other hand, there was concern about bone fusion failure due to the narrow bone graft surface or not performing bone grafting in CMIS for ASD [15]. In this study, bone fusion morphology and bone fusion rate were investigated in CMIS for ASD. This study reported bone fusion rates of 95% and 89% in the lumbar spine and lumbosacral spine, respectively, which were similar to those reported previously. In previous studies, the bone fusion rate in corrective surgery for ASD was reported as 80–95% [9,16,17,18,19]. However, the rate of bone fusion in the thoracic spine without bone grafting was 54%. Few researchers have investigated the bone fusion process and morphology in the thoracic spine without bone grafting in CMIS for ASD. In Group U, type N was significantly lower in preoperative vertebral conditions, and type B was significantly higher in fusion morphology in the thoracic spine and lumbosacral spine. We speculated that the severe degeneration and deformation of the thoracolumbar spine, seen in most patients with spinal deformities, contribute to this result. Moreover, osteophyte formation is vigorous in such patients, where active bridging formation and bone fusion are expected. Especially in the lumbar spine, osteoblasts are induced, and bone fusion is promoted by the release of the annular ligament and osteophytes resulting from the LLIF surgical procedure. Izeki et al. reported that 58% of patients with spinal canal stenosis who underwent minimally invasive surgery using LLIF and PPSs had spontaneous facet fusion after two years postoperatively, and the presence of facet osteoarthrosis was reported as a predictor of bone fusion [20]. In this study, in Group U, type N was significantly lower in the thoracic and lumbosacral vertebrae for preoperative vertebral conditions, and type B was significantly higher for bone fusion morphology. This suggests that most patients in Group U had vertebral body and osteophytic degeneration, that bony bridges are formed between the osteophytes, and that bone fusion may be achieved, which is similar to that reported by Izeki et al. [20].

Although the incidence of screw loosening was significantly higher in Group NU, there was no significant difference in ODI or incidence of postoperative implant-related complications, including PJK and RF. This result reveals two novel discoveries. The first discovery is that spontaneous bone fusion develops if the screw is not loose. The reason is that, without screw loosening, movement between vertebral bodies is reduced, resulting in spontaneous bone fusion in degenerated vertebral bodies and facet joints. Second, the mid-term clinical outcomes (at three years) are not negatively affected, even when spontaneous bone union failure occurs in the thoracic spine. As a result of thoracic bone union failure due to screw loosening, thoracic spine movement remains present. We speculated that this contributes to a smooth transition between instrumented and non-instrumented vertebral bodies and, as a result, a reduction of adjacent segment disease and vertebral fracture. Furthermore, various risk factors have been reported to contribute to the occurrence of PJK [21,22]. Some reports indicate that the preservation of the posterior soft tissue is important for the prevention of PJK [6,8,9]. If the clinical results without bone grafting have no significant differences, it may be more useful to prevent PJK by using PPS to reduce posterior soft tissue detachment and damage. In this study, the rate of spontaneous thoracic spinal bone fusion increased over time in patients without screw loosening; hence, when there is no screw loosening, a further increase in bone fusion is expected in the long-term follow-up. This further suggests that thoracic bone grafting is not necessary. Zou et al. reported that a low HU value was a risk factor for PS loosening [23]. Yao et al. reported that HU could also be a predictor of bony PJK [24]. In the present study, the HU values between the SL and NSL groups had no significant differences, and PSA was concluded as the risk factor for screw loosening, in contrast to previous reports. Furthermore, screw loosening was a risk factor for spontaneous bone fusion failure in the thoracic spine. From this result, we can speculate that increasing the PSA angle is more important for obtaining spontaneous bone fusion than improving bone mineral density (BMD). From the PSA cutoff value of 15 degrees, we can speculate the following two points. The first is that a longer screw can be inserted by tilting the PPS insertion trajectory in the UIV toward the caudal side. This may contribute to the increase in the pull-out strength of the screw and the prevention of screw loosening. Oe et al. reported that longer screws could reduce the stress on the UIV and UIV fracture [25]. Second, the screw trajectory in the caudal direction may contribute to the prevention of pull-out because it is in a different direction from the screw pull-out vector.

### Limitations

This study had several limitations. First, the number of cases was small. Second, BMD has not been evaluated, although it is widely known that BMD has a significant effect on bone fusion. Okuyama et al. demonstrated that, among 52 patients who underwent posterior lumbar interbody fusion, a significantly higher BMD was recorded in patients with bone union than in those with bone union failure [26]. In this study, the HU values had no significant differences between both groups, but a significant difference existed in vertebral conditions and screw loosening, indicating that vertebral conditions and screw loosening impact bone fusion more than BMD. Third, the use of osteoporotic drugs, such as teriparatide and denosumab, has not been evaluated. Some studies have reported that teriparatide administration reduces screw loosening and PJK. Sawakami et al. reported that preoperative teriparatide (TP) administration decreased the chances of screw loosening and subsequent fracture after long bone fusion [27]. Kawabata et al. reported that administering TP reduces the rate of mechanical complications in osteoporotic vertebral fractures following posterior instrumentation [28]. Further, Yagi et al. reported that TP decreased the occurrence of PJK after a long fusion at a 2-year follow-up [29]. In this study, there was no significant difference in HUs and the occurrence of PJK between the NU and U groups and between the SL and NSL groups. Based on this result, osteoporotic drugs may not be useful for bone fusion and preventing PJK. Fourth, the screw length in the UIV was not evaluated. Oe et al. reported that longer screws in the UIV significantly prevented PJF [25]. In this study, the PSA was identified as a risk factor for screw loosening. A larger PSA allowed the insertion of longer screws, and screw length may be another important factor that requires evaluation in the future. Fifth, the proportion of patients with lumbar lordosis was not evaluated. Yilgor et al. reported that the proportion of postoperative spinal alignment affects the occurrence of mechanical complications [30]; hence, the global alignment proportion score should be evaluated in future studies. Seventh, this subject was followed for three years which may be insufficient. Because of the possibility of various complications, it may take more than ten years to conclude that bone grafting is not necessary in the thoracic spine. Therefore, a longer-term investigation is needed.

## 5. Conclusions

We investigated the bone fusion rate and morphology in CMIS using LLIF and PPSs for ASD. A spontaneous bone fusion rate of 54% was obtained in the thoracic spine without a bone graft, while the bone fusion rates in the lumbar spine and lumbosacral spine were 95% and 89%, respectively. The most common bone fusion morphology in the thoracic, lumbar, and lumbosacral spine was type B bone fusion, followed by type P bone fusion in the thoracic spine, while type I bone fusion was the second most common in the lumbar and lumbosacral spine. In the thoracic spine without bone grafting, there was no significant difference in the clinical outcomes and the incidence of implant-related complications between Group NU and Group U. This result clarified that there were no problems with the mid-term (3-year) results of CMIS without bone grafting in the thoracic spine. Risk factors for spontaneous bone fusion failure were post-SVA, screw loosening, and a type N vertebral condition of the thoracic spine.

## Figures and Tables

**Figure 1 medicina-58-00496-f001:**
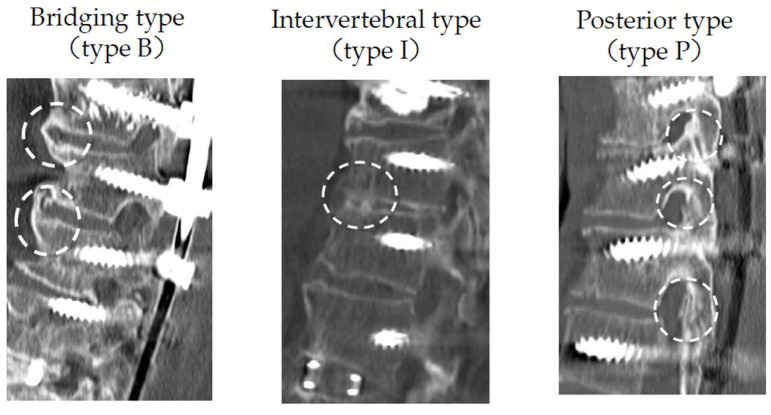
Classifications of bone fusion morphology. Abbreviations: type B, bridging type; type I, intervertebral fusion; type P, posterior fusion type.

**Figure 2 medicina-58-00496-f002:**
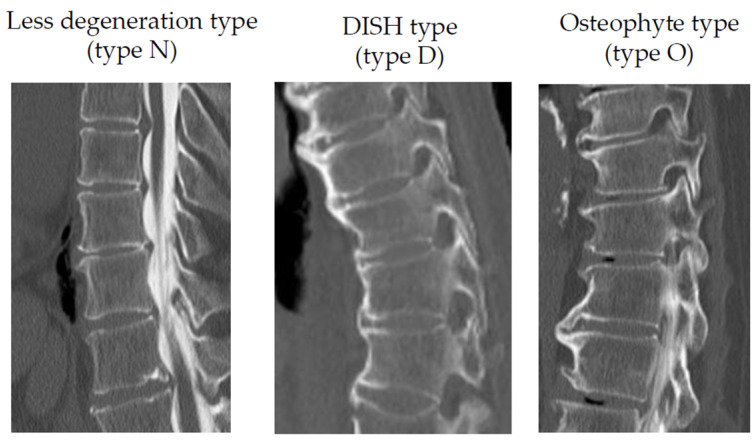
Classifications of vertebral body condition of the thoracic spine. Abbreviations: DISH, diffuse idiopathic skeletal hyperostosis; type N, less degenerative; type D, DISH; type O, osteophyte.

**Figure 3 medicina-58-00496-f003:**
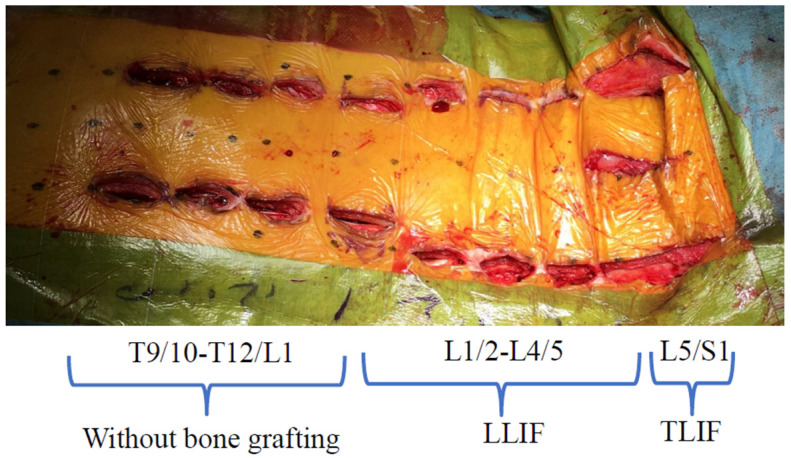
Intraoperative photo. Abbreviations: LLIF, lateral lumbar interbody fusion; TLIF, transforaminal lumbar interbody fusion.

**Figure 4 medicina-58-00496-f004:**
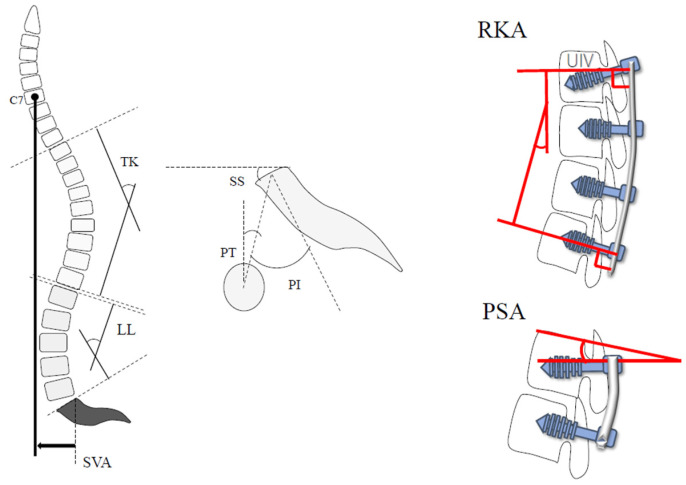
Illustration of various spinopelvic parameters and implant-related parameters. Abbreviations: LL, lumbar lordosis; TK, thoracic kyphosis; PI, pelvic incidence; PT, pelvic tilt; SS, sacral slope; RKA, rod kyphotic angle from the upper instrumented vertebra (UIV) to L1; PSA, pedicle screw angle in UIV.

**Figure 5 medicina-58-00496-f005:**
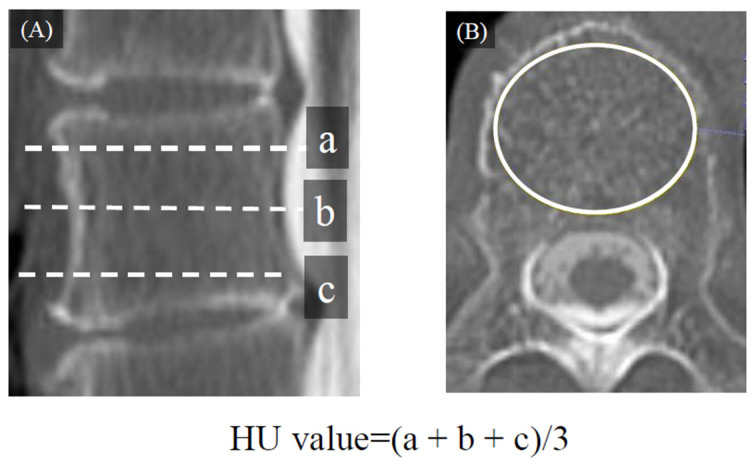
Measurement of Hounsfield unit value of UIV. (**A**): preoperative CT sagittal plane in UIV, (**B**): preoperative CT axial plane in UIV. Abbrevia-tions: HU, Hounsfield unit.

**Figure 6 medicina-58-00496-f006:**
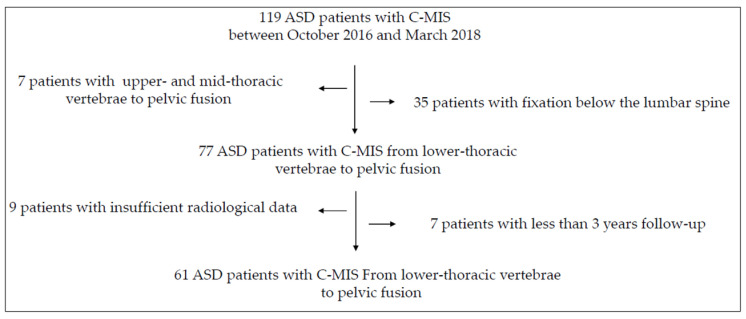
Patient selection diagram.

**Figure 7 medicina-58-00496-f007:**
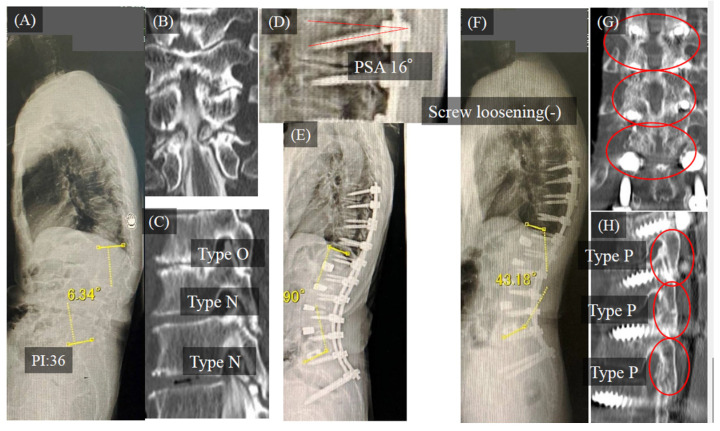
Case study. (**A)**: preoperative standing whole spine lateral view, (**B**): preoperative CT coronal plane, (**C**): preoperative CT sagittal plane, (**D**): postoperative X-ray, (**E**): postoperative standing whole spine lateral view, (**F**): postoperative standing whole spine lateral view two years after surgery, (**G**): postoperative CT coronal plane two years after surgery, (**H)**: postoperative CT sagittal plane two years after surgery. Abbreviations: CT, computed tomography.

**Table 1 medicina-58-00496-t001:** Demographic and baseline data.

Parameters	Findings (n = 61)	Number of Cases
Age (years)	73.2 ± 7.3 (52–82)	
Gender (male: female)	15:46	
Duration of follow-up (month)	49.3 ± 5.5 (38–58)	
Number of fused levels (segment)	10.3 ± 0.5 (10–13)	
Number of LLIF (segment)	4.0 ± 0.5 (3–5)	
Number of corpectomy cases	3%	2 cases
UIV (case)		
T9	30%	18 cases
T10	70%	43 cases
Operative time (min)		
Anterior	128.0 ± 30.4 (78–218)	
Posterior	255.0 ± 44.1 (171–387)	
Total	381.5 ± 59.6 (289–541)	
Estimated blood loss (mL)		
Anterior	103.4 ± 143.1 (5–605)	
Posterior	612.7 ± 301.7 (79–1530)	
Total	716.1 ± 358.8 (123–1610)	
Complications (%)		
Screw loosening	46%	28 cases
UIV and UIV-1	13%	9 cases
UIV	33%	20 cases
PJK	20%	12 cases
RF		
Thoracic	0%	0 cases
Lumbar	15%	9 cases
Lumbosacral	13%	8 cases

Values are presented as mean ± standard deviation. Abbreviations: PJK, proximal junctional kyphosis; RF, rod fracture.

**Table 2 medicina-58-00496-t002:** Bone fusion rates and morphology after three postoperative years.

		Thoracic (n = 191, 61 Cases)		Lumbar (n = 242)	Lumbosacral (n = 61)
Bone fusion rate		54% (104/191, 33 cases)		95% (231/242)	89% (54/61)
		New bone fusion	42% (81/191)		
		Existing bone fusion	12% (23/191)		
Fusion morphology	Type B	54% (56/104, 18 cases)		72% (168/231)B + I 37%, B + P 30%	74% (40/54)
	Type I	10% (10/104, 3 cases)		27% (63/231)	19% (10/54)
	Type P	36% (38/104, 12 cases)		0% (0/231)	6% (3/54)

**Table 3 medicina-58-00496-t003:** Bone fusion rate by vertebral conditions in the thoracic spine.

		Type N (n = 40 Cases 124 Levels)	Type D (n = 11 Cases 48 Levels)	Type O (n = 10 Cases 31 Levels)	*p*-Value	
					Type N vs. Type D	Type N vs. Type O
Fusion rate	(%)	35%	100%	80%	<0.001 *	0.013 *
	cases	14/40	11/11	8/10		
	levels	46/124	36/36	26/31		

Fisher’s exact test. * Indicates statistically significant values.

**Table 4 medicina-58-00496-t004:** Comparison of demographic data between Group U and Group NU.

			Group U (n = 33)	Group NU (n = 28)	*p*-Value
Age			73.5 ± 7.3	72.8 ± 7.4	0.346
Sex (male: female)			10:23	5:23	0.207
vertebral condition	type D (cases)		11/33	0/28	<0.001 *
	type O (cases)		8/33	2/28	0.071
	type N (cases)		14/33	26/28	<0.001 *
complications	screw loosening (cases)		8/33	20/28	<0.001 *
	PJK (cases)		5/33	7/28	0.466
	RF (case)	thoracic	0/33	0/28	1.000
		lumbar	3/33	5/28	0.260
		lumbosacral	2/33	6/28	0.264
		total	5/33	11/33	0.032 *
revision rate	RF lumbosacral (cases)		5/33	9/28	0.102
	PJK (cases)		3/33	5/28	0.264
	Total (cases)		7/33	10/28	0.165
HU			125.1 ± 43.1	134.7 ± 50.1	0.212

Fisher’s exact test. * Indicates statistically significant values. Abbreviations: Group U, union group; Group NU, nonunion group.

**Table 5 medicina-58-00496-t005:** Spinopelvic parameters (Group U vs. Group NU).

	Group U (n = 33)	Group NU (n = 28)	*p*-Value
Pre-PI (°)	46.1 ± 12.9	48.0 ± 12.3	0.310
Post-PI (°)	48.2 ± 12.5	49.2 ± 11.9	0.351
Pre-LL (°)	13.5 ± 13.8	6.8 ± 18.7	0.081
Post-LL (°)	47.4 ± 12.2	47.2 ± 10.0	0.473
Pre-PI-LL (°)	32.6 ± 15.7	41.1 ± 20.1	0.054
post-PI-LL (°)	−1.2 ± 11.6	0.7 ± 11.1	0.269
Pre-PT (°)	28.7 ± 10.6	32.2 ± 21.3	0.149
Post-PT (°)	16.6 ± 8.0	15.9 ± 11.6	0.399
Pre-TK (°)	22.8 ± 14.4	13.6 ± 13.4	0.006 *
Post-TK (°)	38.9 ± 10.0	36.1 ± 8.7	0.140
Pre-SVA (mm)	82.3 ± 45.1	107.9 ± 40.1	0.076
Post-SVA (mm)	24.4 ± 28.5	37.8 ± 28.8	0.037 *
PSA (°)	10.9 ± 8.2	6.6 ± 8.8	0.025 *
RKA (°)	20.4 ± 6.9	20.4 ± 7.9	0.493

Values are presented as mean ± standard deviation. Student’s *t*-test * indicates statistically significant values. Abbreviations: SVA, sagittal vertical axis.

**Table 6 medicina-58-00496-t006:** Risk factors of bone fusion failure in the thoracic spine (logistic regression analysis).

	Odds Ratio	95% CI	*p*-Value
Pre-TK	0.979	0.927–1.029	0.409
Post-SVA	1.028	1.003–1.059	0.022 *
PSA	0.989	0.899–1.087	0.830
Vertebral condition type N	23.812	3.443–329.921	<0.001 *
screw loosening	6.177	1.356–35.964	0.017 *

* Indicates statistically significant values. The values for postoperative SVA, type N vertebral body condition, and PS loosening were significantly different (*p* < 0.05). Abbreviations: CI, confidence interval.

**Table 7 medicina-58-00496-t007:** Comparison of postoperative outcomes between Group SL and Group NSL.

			Group NSL (n = 33)	Group SL (n = 28)	*p*-Value
Age			72.5 ± 8.5	74.1 ± 5.7	0.207
Gender (male: female)			9:24	5:23	0.207
Vertebral condition	type D (cases)		9/33	2/28	0.028 *
	type O (cases)		7/33	3/28	0.568
	type N (cases)		17/33	23/28	0.001 *
Complications	PJK (cases)		12% (4/33)	29% (8/28)	0.118
	RF (case)	thoracic	0/33	0/28	1.000
		lumbar	2/33	6/28	0.082
		lumbosacral	4/33	4/28	0.549
		Total	6/33	10/28	0.104
Revision rate	RF (lumbosacral) (cases)		5/33	10/28	0.059
	PJK (cases)		3/33	5/28	0.264
	Total (cases)		7/33	10/28	0.165
Bone fusion rate in thoracic spine			76% (25/33) (78/102) segments)	29% (8/28) (26/89 segments)	<0.001 *
HU			138.1 ± 40.7	121.1 ± 51.3	0.078

Fisher’s exact test. Student’s *t*-test. * Indicates statistically significant values. Abbreviations: Group SL, pedicle screw loosening group; Group NSL, non-loosening group.

**Table 8 medicina-58-00496-t008:** Spinopelvic parameters (SL group vs. NSL group). Values are presented as mean ± standard deviation.

	Group NSL (n = 33)	Group SL (n = 28)	*p*-Value
Pre-PI (°)	44.7 ± 12.8	51.0 ± 11.1	0.023 *
Post-PI (°)	45.3 ± 11.9	51.2 ± 10.7	0.038 *
Pre-LL (°)	5.2 ± 17.5	12.9 ± 16.9	0.04
Post-LL (°)	47.4 ± 11.1	47.2 ± 11.1	0.476
Pre-PI-LL (°)	35.9 ± 19.1	38.0 ± 18.0	0.344
post-PI-LL (°)	−1.0 ± 11.1	3.0 ± 11.0	0.078
Pre-PT (°)	30.3 ± 12.8	30.7 ± 10.3	0.456
Post-PT (°)	14.2 ± 10.4	18.6 ± 8.9	0.064
Pre-TK (°)	22.3 ± 15.4	14.2 ± 12.6	0.014 *
Post-TK (°)	38.8 ± 9.6	36.1 ± 9.2	0.165
Pre-SVA (mm)	93.7 ± 61.3	106.1 ± 51.7	0.201
Post-SVA (mm)	27.5 ± 29.4	31.9 ± 30.7	0.311
PSA (°)	12.1 ± 7.9	5.1 ± 7.8	<0.001 *
RKA (°)	20.1 ± 7.1	20.7 ± 7.7	0.055

* Indicates statistically significant values (Student’s *t*-test).

**Table 9 medicina-58-00496-t009:** Risk factors of PS loosening in UIV (logistic regression analysis).

	Odds Ratio	95%CI	*p*-Value
PI	1.041	0.985–1.108	0.147
Pre-LL	1.040	0.995–1.093	0.079
Pre-TK	0.978	0.927–1.026	0.381
PSA	1.115	1.025–1.231	0.009 *
Vertebral condition type N	4.107	0.967–20.666	0.055

* Indicates statistically significant values. The PSAs were significantly different (*p* < 0.05).

**Table 10 medicina-58-00496-t010:** ROC curve analysis.

	Cutoff Value	Sensitivity	Specificity	AUC
PSA	15.3°	1.00	0.425	0.731

The ROC analysis of PS loosening revealed 15.3° and 0.731 as the cutoff value and AUC, respectively. Abbreviations: ROC, receiver operating characteristic; AUC, area under the curve.

## Data Availability

The data used in this study are available upon reasonable request from the corresponding author.

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
