# Peer review of "Bone Fusion Morphology after Circumferential Minimally Invasive Spine Surgery Using Lateral Lumbar Interbody Fusion and Percutaneous Pedicle Screws without Bone Grafting in the Thoracic Spine: A Retrospective Study"

_medicina, 2022, doi:10.3390/medicina58040496_

Round 1

Reviewer 1 Report

Comments

Authors investigated retrospectively that the way of fusion at thoracic spine after corrective surgery for ASD with their case series(n=61). They concluded the difference of spinopelvic parameter, ODI and rate of PJK, RF were not found between fusion group and non-fusion group. But screw loosening was found significantly in the non-fusion group and less degenerated thoracic spine and screw loosing were confirmed as the risk of fusion failure at thoracic spine within the instrumented vertebra. Finally, they suggested that the bone grafting might be unnecessary around the upper instrumented vertebra, because the spontaneous bone fusion failure at thoracic spine within the instrumented vertebra might not be affect the clinical results after the corrective surgery for ASD.

Q1

The fact that this report is three years evaluation have to be emphasized in limitation section or title. Because ten years might be required for the conclusion that the bone graft around the upper instrumented vertebra is unnecessary.

Q2

Are these many separate skin incisions really important?  I wonder if these things can make the correction difficult in ASD?

Q3

In the line 172, 13% of screw loosening was found in UIV-1? or UIV+1?  

Q4

Could you let me know your thoughts and speculations about PSA is large in union group or non-screw loosening group. Could you comment it in your manuscript?

Q5

Please delete the private information in figure 6.

Q6

Izeki said that the facet osteoarthrosis is a predictor for the failure of spinal fusion. But your results showed that degrative changes allows good fusion rate. Even though these two are not similar results, you said in the discussion section that these results look like the same.

Conclusion

This paper is well documented about the fusion status around the upper instrumented vertebra with various parameters. Some revisions that I mentioned above is required.

Author Response

Our point-by-point response to the reviewers’ comments and suggestions is listed below:

We thank you for taking the time and effort necessary to review our manuscript and provide us with these valuable comments and suggestions. Accordingly, we revised our manuscript and made changes to it. Please note that the yellow highlighted parts represent our responses to the comments.

Review 1

Comments

Authors investigated retrospectively that the way of fusion at thoracic spine after corrective surgery for ASD with their case series(n=61). They concluded the difference of spinopelvic parameter, ODI and rate of PJK, RF were not found between fusion group and non-fusion group. But screw loosening was found significantly in the non-fusion group and less degenerated thoracic spine and screw loosing were confirmed as the risk of fusion failure at thoracic spine within the instrumented vertebra. Finally, they suggested that the bone grafting might be unnecessary around the upper instrumented vertebra, because the spontaneous bone fusion failure at thoracic spine within the instrumented vertebra might not be affect the clinical results after the corrective surgery for ASD.

Q1

The fact that this report is three years evaluation have to be emphasized in limitation section or title. Because ten years might be required for the conclusion that the bone graft around the upper instrumented vertebra is unnecessary.

Response: We thank you for this comment. As you pointed out, a limitation of our study is that the subject was only followed for three years. We added that concluding that bone grafting is not needed in the thoracic spine may take more than ten years. (Lines 363-366).

Q2

Are these many separate skin incisions really important?  I wonder if these things can make the correction difficult in ASD?

Response: We thank you for this comment. Flexibility is gained due to intervertebral release at L1 / 2 ~ L4 / 5 and, in some cases, T11 / 12 / L1 at the LLIF. Sufficient correction is achieved by gradually applying PPS inserted into each vertebral body to the rod as a whole (translation).

Q3

In the line 172, 13% of screw loosening was found in UIV-1? or UIV+1?  

Response: We thank you for this comment. In 13% of all patients, UIV and UIV-1 loosened. In an additional 33%, only the UIV loosened (Table 1).

Q4

Could you let me know your thoughts and speculations about PSA is large in union group or non-screw loosening group. Could you comment it in your manuscript?

Response: We thank you for this comment.

screw loosening was a risk factor for spontaneous bone fusion failure in the thoracic spine. From this result, we can speculate that increasing the PSA angle is more important for obtaining spontaneous bone fusion than improving bone mineral density (BMD). From the PSA cutoff value of 15 degrees, we can speculate the following two points. The first is that a longer screw can be inserted by tilting the PPS insertion trajectory in the UIV toward the caudal side. This may contribute to the increase in the pull-out strength of the screw and the prevention of screw loosening. Oe et al. reported that longer screws could reduce the stress on the UIV and UIV fracture [22]. Second, the screw trajectory in the caudal direction may contribute to the prevention of pull-out because it is in a different direction from the screw pull-out vector.

We have added the above content to our discussion. (line 324-334)

  1. Oe S, Narita K, Hasegawa K, et al. Longer screws can reduce the stress on the upper instrumented vertebra with long spinal fusion surgery: A finite element analysis study. Global Spine J 2021;21925682211018467. [PMID: 34002639 DOI: 10.1177/21925682211018467]

Please delete the private information in figure 6.

Response: We thank you for this comment. We deleted the private information in figure 6.

Q6

Izeki said that the facet osteoarthrosis is a predictor for the failure of spinal fusion. But your results showed that degrative changes allows good fusion rate. Even though these two are not similar results, you said in the discussion section that these results look like the same.

Response: We thank you for this comment. We included it in error and have corrected it as follows (Lines 292–295):

Izeki et al. reported that 58% of patients with spinal canal stenosis who underwent minimally invasive surgery using LLIF and PPSs had spontaneous facet fusion after 2 years postoperatively, and the presence of facet osteoarthrosis was reported as a predictor of bone fusion [17].

Conclusion

This paper is well documented about the fusion status around the upper instrumented vertebra with various parameters. Some revisions that I mentioned above is required.

Reviewer 2 Report

This retrospective study is an interesting study examining the rate and morphology of bone fusion in the thoracic and lumbosacral spine, the incidence of implant-related complications associated with bone fusion failure, and the risk factors for thoracic spontaneous bone fusion failure in CMIS using LLIF and PPS without bone graft in the thoracic spine.

The paper is well designed and written.

However, I think that there are several concerns that should be made before publication.

Abstract

There is a discrepancy between the objectives and the results, so please correct one or the other.

Author conclusion is an overestimation or SPIN, as only three-year results are available and the long-term results are not known.

“Therefore, bone grafting of the thoracic spine may not be necessary.”

In Abstracts and Manuscripts, authors should unify their conclusions.

2)Materials and Methods

Please include an additional Flow diagram of eligible patients to reduce the risk of selection bias.

3)Discussion

>Line291 The first discovery is that spontaneous bone fusion is easy to obtain if the screw is not loose.

It would be easier to understand if the higher levels and sites were explained in a more complementary way.

4)Conclusion

>We investigated the bone fusion rate and morphology in CMIS using LLIF and PPSs for ASD.

For bone fusion rate and morphology, add some more supplementary information on the site

Author Response

Our point-by-point response to the reviewers’ comments and suggestions is listed below:

We thank you for taking the time and effort necessary to review our manuscript and provide us with these valuable comments and suggestions. Accordingly, we revised our manuscript and made changes to it. Please note that the yellow highlighted parts represent our responses to the comments.

Review2

Open Review

This retrospective study is an interesting study examining the rate and morphology of bone fusion in the thoracic and lumbosacral spine, the incidence of implant-related complications associated with bone fusion failure, and the risk factors for thoracic spontaneous bone fusion failure in CMIS using LLIF and PPS without bone graft in the thoracic spine.

The paper is well designed and written.

However, I think that there are several concerns that should be made before publication.

Abstract

There is a discrepancy between the objectives and the results, so please correct one or the other.

Author conclusion is an overestimation or SPIN, as only three-year results are available and the long-term results are not known.

“Therefore, bone grafting of the thoracic spine may not be necessary.”

In Abstracts and Manuscripts, authors should unify their conclusions.

Response: We thank you for this comment. As you pointed out, this study presents the clinical results of this procedure for ASD (adult spinal deformity) three years after surgery, and its effect on long-term clinical results is unknown. Therefore, I deleted the sentence " Bone grafting of the thoracic spine may not be necessary." in the abstract and replaced it with "there is no effect on clinical results with the mid-term (3-year) results of CMIS without bone grafting in the thoracic spine." (Lines 39–41).

2)Materials and Methods

Please include an additional Flow diagram of eligible patients to reduce the risk of selection bias.

Response: We thank you for this comment. We have created a flow chart showing the patient selection process for this study (Figure 6, Line 166). We have included additional details about the selection process in the text (Lines 160–163).

3)Discussion

 The first discovery is that spontaneous bone fusion develops when if the screw is not loose (Line 305).

It would be easier to understand if the higher levels and sites were explained in a more complementary way.

Thank you for your comment. However, we are not clear what you are asking about. Can you provide additional information so we can appropriately address your concerns?

4)Conclusion

>We investigated the bone fusion rate and morphology in CMIS using LLIF and PPSs for ASD.

For bone fusion rate and morphology, add some more supplementary information on the site

Thank you for your comment. We have added the bone fusion rate and bone fusion morphology in the thoracic spine (Table 2, Line 188, 372-375).

Reviewer 3 Report

The study presented by Ishihara and co-workers investigates the process of thoracic and lumbosacral bone fusion in patients with adult spinal deformity (ASD) who underwent circumferential minimally invasive spine surgery (CMIS) by lateral lumbar interbody fusion and percutaneous pedicle screws (PPSs). They evaluate the rate and morphology of bone fusion and rod fracture rate in the thoracic and lumbosacral vertebrae by means of a retrospective study that includes 61 patients with spinal deformities (46 women and 15 men) who underwent CMIS with LLIF and PPSs at their hospital after 2016 and completed a 3-year postoperative follow-up diving patients into a thoracic spine spontaneous bone fusion group and a bone fusion failure group. They found that screw loosening in the thoracic vertebrae was significantly higher in the bone fusion failure group, while no significant differences were observed in the spinopelvic parameters, Oswestry Disability Index (ODI), and rate of proximal junctional kyphosis and rod fractures. On the basis of these results, they concluded that bone grafting of the thoracic spine may not be necessary since spontaneous bone fusion is likely to be obtained without screw loosening, and even if bone fusion is not obtained, there is no effect on clinical results.

The paper is interesting from a medical point of view and the manuscript contains sufficient noteworthy information to justify its publication. The subject is significant and concisely stated, the authors have obtained many interesting data, the statistical analysis been performed appropriately and the interpretation of the results are justified. I have only one minor point to improve the manuscript: adding a table of abbreviations could be very helpful because the extensive use of abbreviations makes sometimes the paper a bit difficult to follow.

Author Response

Our point-by-point response to the reviewers’ comments and suggestions is listed below:

We thank you for taking the time and effort necessary to review our manuscript and provide us with these valuable comments and suggestions. Accordingly, we revised our manuscript and made changes to it. Please note that the yellow highlighted parts represent our responses to the comments.

Reviewer3

Open Review

The study presented by Ishihara and co-workers investigates the process of thoracic and lumbosacral bone fusion in patients with adult spinal deformity (ASD) who underwent circumferential minimally invasive spine surgery (CMIS) by lateral lumbar interbody fusion and percutaneous pedicle screws (PPSs). They evaluate the rate and morphology of bone fusion and rod fracture rate in the thoracic and lumbosacral vertebrae by means of a retrospective study that includes 61 patients with spinal deformities (46 women and 15 men) who underwent CMIS with LLIF and PPSs at their hospital after 2016 and completed a 3-year postoperative follow-up diving patients into a thoracic spine spontaneous bone fusion group and a bone fusion failure group. They found that screw loosening in the thoracic vertebrae was significantly higher in the bone fusion failure group, while no significant differences were observed in the spinopelvic parameters, Oswestry Disability Index (ODI), and rate of proximal junctional kyphosis and rod fractures. On the basis of these results, they concluded that bone grafting of the thoracic spine may not be necessary since spontaneous bone fusion is likely to be obtained without screw loosening, and even if bone fusion is not obtained, there is no effect on clinical results.

The paper is interesting from a medical point of view and the manuscript contains sufficient noteworthy information to justify its publication. The subject is significant and concisely stated, the authors have obtained many interesting data, the statistical analysis been performed appropriately and the interpretation of the results are justified. I have only one minor point to improve the manuscript: adding a table of abbreviations could be very helpful because the extensive use of abbreviations makes sometimes the paper a bit difficult to follow.

Round 2

Reviewer 1 Report

This manuscript is well revised and I think it is good enough for the acceptance.

Author Response

We thank you for your comment.